# Weaving Social Connectivity into the Community Fabric: Exploring Older Adult’s Relationships to Technology and Place

**DOI:** 10.3390/ijerph19148500

**Published:** 2022-07-12

**Authors:** Belinda Paulovich, Sonja Pedell, Erica Tandori, Jeanie Beh

**Affiliations:** 1Centre for Design Innovation, Swinburne University of Technology, John St., Hawthorn, VIC 3122, Australia; spedell@swin.edu.au (S.P.); jebeh@swin.edu.au (J.B.); 2School of Biomedical Sciences, Monash University, Wellington Rd., Clayton, VIC 3800, Australia; erica.tandori@monash.edu.au

**Keywords:** aging, place, community, social connectivity, technology, wellbeing

## Abstract

The wellbeing of older adults is positively impacted by ease of access to social networks and opportunities. In urban fringe communities, longer geographical distances, combined with mobility and health issues, can lead to decreased access to community life. Technology can facilitate socialisation opportunities for older adults living in more geographically isolated locations, but we need to work with communities to better understand how technology can fit into their *existing* social tapestry and community infrastructure. We conducted an explorative, qualitative study consisting of in-depth semi-structured interviews with members of an urban fringe community (n = 2), and a community mapping focus group (n = 14). Transcripts and mapping materials were analysed thematically, and also the method’s suitability explored. The community mapping proved suitable to uncover the complexity of technology use to support social connectivity. We found that while technology was perceived as valuable by our participants, there were also significant fears and concerns surrounding its use related to the abstract concept of online friendship and the steep learning curve required to master some platforms. Inclusive communities connected by technology require tailored and customised community-led technology initiatives in order to accommodate for the unique social and geographical contexts in which they live. We outline the next steps for future research on technology-supported social connectivity within urban fringe communities.

## 1. Introduction

While numerous technology-based interventions have been developed to prevent isolation and enhance social connection, there is little evidence around how effective they are for older people [1]. This is of particular relevance in communities on the urban fringe where longer geographic distances, combined with mobility issues of older people and fewer public services available, can lead to isolation and decrease their access to community life—consequently impacting overall wellbeing.

The term ‘urban fringe’ was mentioned in 1937 when T.L. Smith used it to signify ‘the built-up area just outside the corporate limits of the city’ [2]. Pryor’s definition of urban fringe is “*the zone of transition in land use, social and demographic characteristics, lying between the continuously built up urban and suburban areas of the central city, and the rural hinterland, characterised by the almost complete absence of nonfarm dwellings, occupations and land use*” [2] (p. 5). Since then, the term has been used in the academic literature for the transition zone between city and countryside. Urban fringe is also referred to as peri-urban by some literature [3]. Urban fringe areas are generally located in close proximity to transport infrastructures, farmlands, and urban subdivisions [4].

Urban fringe areas are often perceived as serene, rustic places populated by people who value self-sufficiency and community spirit [5]. While we recognise that many older people living in such communities have deeply embedded connections to people and place, we know that relationships amongst community members are complex [6] and that a one-size-fits-all description of social values only serves to generalise or reduce what is a very rich tapestry of interwoven elements. This exploratory, qualitative study unravels the detail of social connections amongst members of a small urban fringe community in order to better understand the complexities of their networks and social circles, and the role of technology in facilitating these. With increasing numbers of people choosing to live outside of urban areas as they grow older [7], it is of great importance that we investigate how socialisation opportunities can be improved for older adults living in more geographically isolated locations.

### 1.1. Older Adults and Social Connectivity

The wellbeing of older adults is negatively impacted by the absence of positive social relationships in their life, and the extent to which they feel isolated [8,9,10,11]. High levels of perceived social isolation (which can be geographical and/or emotional) have been shown to be particularly damaging to older adults’ wellbeing over time) [12] (p. 686). Critical to wellbeing is the notion of mobility, with older adults reporting that it is part of their sense of self and feeling whole. With declining movement, assisted mobility becomes fundamental to wellbeing, and adaptability is required to move forward [13,14].

Numerous initiatives have been developed with the aim of reducing social isolation [15] and loneliness [16] in older people. Many of the initiatives documented in the literature involve a face-to-face component, indicating that proximity to people is a meaningful factor in the way that older adults establish and maintain relationships. For example, a singing community group (Golden Oldies) was studied [17]. Participation in the group was shown to reduce social isolation and increase social contact, to be a source of therapy, and provided a new lease of life. Social connections appeared to be a significantly important thread that contributed to perceived benefits in older adults’ quality of life [17,18,19]. Face-to-face initiatives with a primary motivation (such as exercising) appear to have a significant social benefit for participants, even though this is not the primary aim, such as in the Preventing Loss of Independence through Exercise program (PLIE) [20].

Since conducting this study, the way in which people interact with each other has been impacted by COVID-19. Throughout 2020, 2021 and 2022, the Victorian State Government implemented a number of measures to curb the spread of the virus. These included social distancing, stay-at-home orders, and the wearing of masks in both indoor and outdoor spaces. As the pandemic continues, we need to consider that face-to-face interaction may remain limited, and that people of all ages may experience loneliness because of lockdowns and other social distancing measures [21], placing even more importance on how technology can be used to foster social connections virtually.

### 1.2. The Supporting Role of Technology

Technology can play an important supporting role in the social wellbeing of older adults. This sentiment is summarised by William Chopik who stated, ‘Close relationships are a large determinant of physical health and wellbeing, and technology has the potential to cultivate successful relationships amongst older adults’ [22] (p. 551).

Technology is commonly used by older adults to maintain family and social connections and to access information on health and routine activities [23]. Patterns of technology use in this cohort are varied and are influenced by factors including age, education, attitudes, personality and marital status [23]. However, older adults consistently perceive four factors as critical to their Internet use: social connection, self-efficacy, the need to seek financial information, and the need to seek health information [24]. In later life, social goals take priority and older adults might be more inclined to use technology if the social benefits (rather than the informational benefits) are highlighted [25].

Recognising this, many researchers have developed technological tools and programs to help older adults stay connected. For example, ‘InTouch’ was developed in response to the communication needs of people in isolated environments [9], while an 8-week Wii Bowling program was shown to increase social connectedness and reduce loneliness in participants [26]. ‘PictureFrame’ is a novel social technology that was developed to allow older adults living at home to share photos and messages with caregivers. It enabled caregivers to monitor wellbeing without positioning older adults as ‘the subjects of care’ [27] (p. 270).

While many programs exist, the literature documents numerous challenges and barriers to technology adoption by older adults. Technology can make a big difference in helping older people feel socially connected [28], but it rarely addresses their needs [29]. For example, older people express concerns about the amount of time required to engage, the loss of deeper communication, content irrelevance, and privacy [30]. Research results also confirm concerns around the ‘digital divide’ (strong disparities in technology use between younger and older generations), with many older adults resisting participation in mainstream technologies used by younger members of their social network [22,30]. There is a call for the design of ‘bridging technologies’ that meet the accessibility, cultural practice needs, and preferences of multiple generations [30], as well as multiple domains [31]. However, even when technologies have been specifically designed for older adults, they may choose not to engage with them [32]. This indicates that a ‘one-size-fits-all’ approach to socialisation opportunities for a heterogenous demographic, such as older adults, is not valid [32]. We are primarily interested in how technologies can be used and deployed within communities, and how these technologies can be improved, adopted, and innovated by communities of older adults [33]. The research question underpinning this exploratory study is: *how can technology fit into the existing social network and community infrastructure to better support social connectivity?* Our aims are (i) to examine the complex relationships between community members’ social connections, place and technology, and (ii) to determine suitable methods for exploring these existing relationships. Our reasons for examining the opportunities for technology use by older adults from a community perspective are because technology has the potential to better connect the individual to their community and place. As research shows that social connectivity is one of the main purposes for older adults’ use of technology (e.g., [18]), we expect that this approach will promote and enable a more age appropriate and age friendly engagement.

## 2. Methods

To better understand how technology might fit into the existing tapestry of an urban fringe community, this explorative, qualitative study was conducted in a small town approximately 34 km from Melbourne, Victoria. This community was chosen due to its close proximity to Melbourne, and its small population (3500 people). The Australian Institute of Health and Welfare Rural, Remote and Metropolitan Areas (RRMA) classification defines the selected community as a rural area with an urban centre population of less than 10,000 people [34]. With the study participants, we co-created the pseudonym, “Heritage Village”, to identify the community with a meaningful descriptor whilst maintaining participant anonymity. Approximately 12% of the population in Heritage Village is aged over 65 years [35], and according to the 2016 Census, 91.5% of occupied private dwellings had internet access [35]. Heritage Village has good health and medical services, convenient access to transport infrastructure, and a number of community sporting, environmental, wellbeing and social groups. Study participants (n = 16) were recruited through community contacts and snowball sampling, a method by which participants recruit other participants [36]. They were well-connected community members, aged 59 years or over, who were involved in several social groups and community activism efforts. A high proportion of males took part in the study (n = 14) due to strong recruitment from a men-only social group. As is common in small communities outside the city, most participants were born in Australia or were English native speakers (as some participants were born in the UK). Only two participants did not speak English as their first language.

It is important to note that the research is limited by geography and that the findings from a small case study of one small town in Victoria cannot be generalised to another population [37] (p. 139); [38] (p. 15). Furthermore, the small number of participants and skewed gender sample may limit generalisability, though we do intend to increase the scope of the study once the exploratory study is complete. Other limitations stem from the fact that this research has attracted participants that were already well connected and able to participate.

While transferability of data was limited, this study contributes to other measures of quality research such as credibility, dependability and confirmability [39]. In terms of credibility, we gained confidence though the rich study results that our method of community mapping created. High quality is necessary when exploring topics as complex and nuanced as social connectivity in urban fringe communities. We argue that credibility is also high, due to the rich engagement with the material. The method has been described in enough detail to enable other researchers to repeat its use, thereby fulfilling the minimum requirements on dependability, according to [40]. The analysis section demonstrates how findings emerge from the data, and the quality measure of confirmability [39].

### 2.1. Stage One: Community Interviews

An in-depth semi-structured interview (2 h and 5 min) was conducted with community members (n = 2 female) to gather personal stories about connection to people and place, and personal opinions and experiences of technology use. We wanted to find out about the participants’ lives, the frequency and quality of their social interactions, their motivational goals, and their emotions around both social interactions and technology. The following list of questions was used to guide the discussions:Have you lived here a long time/how did you come to live here?What is it like to live here?How would you describe your local community?How do you maintain social connections with people inside/outside of the community?How do you feel about social technology like mobile phones, Facebook, Skype?

In this initial stage, we realised that while we learned a lot about technology use, we did not really understand how it tied in with the community and place, or how technology use could be tailored more specifically to community needs. Hence, we continued the research in a larger natural group setting. A changed participant setting is not unusual when doing research in communities, and is handled through formative sampling where questions and format can change based on growing insights [41].

### 2.2. Stage Two: Community Mapping Focus Group

Building on information about the strength, frequency and type of social connections that emerged during the stage one interviews, a focus group (2 h) was conducted with community members (n = 14 male) which concentrated on the same questions, but in the context of community mapping. Participants were asked to map their physical and virtual social connections using a range of creative materials and tools (paper, markers and post-it notes).

The instructions provided to participants were as follows:Draw a map of your community including the places that are important to you and the places you visit most often.Using sticky notes, write down how often you visit each place you have included on your map (e.g., once a day, once a week, once a month, once a year).Are there places you visit regularly that are outside of your community? Using sticky notes make a list of these locations and write down how often you visit them.Are there people you contact regularly by technological means (e.g., phone calls or video chats)? Using sticky notes make a list of these people (e.g., friend, daughter, cousin) and where they live (suburb or city), and write down how often you speak with them.

This activity choice was justified through the learnings from stage one. We expected it would be easier and more insightful to talk about community and how their activities manifest in places through visual means. We also wanted to learn about any technology used specifically to maintain the social fabric of the community.

It was decided not to split this group into smaller focus groups as there were two facilitators present who were able to instruct and look after the whole group in their natural group constellation. This enabled the research team to also observe some of their social interactions, such as discussing content of their maps with people sitting next to them. The group demonstrated a level of engagement and enjoyment, as they joked and laughed, pointing to specific aspects of their maps during the activity.

The stage one interviews were audiotaped, transcribed verbatim and analysed using a thematic, interpretive approach [42]. Maps generated by participants in stage two were compiled into one overarching map that acted as a visual representation of the community fabric. It highlighted the physical and virtual connections between people and places, ultimately revealing the extent and complexity of the social network, both within and beyond the physical boundaries of the community (see Figure 1). Locations identified on the participant maps were entered into *Google My Maps* by the research team using a colour coding system (places within Victoria were red, places interstate were yellow, and overseas locations were purple). Within *Google My Maps* (see Figure 2), each participant map has its own layer and associated data table enabling the researcher to see individual community rims and the collective community rim (‘community rim’ is a term that emerged from a participant map that refers to the boundaries or parameters that define a particular community).

## 3. Findings

Four overarching themes were identified across the community interviews and mapping focus group in regard to participants’ experiences and feelings about their technology use, which are detailed in the following sections.

### 3.1. Hesitation about Online Friendships and Social Technology

Participants raised concerns about young people’s engagement with social technology, likening their use to an addiction. They were particularly worried about the immediacy and speed of communication that is facilitated by technology, and suggested that this is creating an unrealistic desire for instant validation amongst the younger generation. In terms of their own engagement with social technology, older adult participants in our study were hesitant about the notion of ‘online friendship’, and suggested that friendship through this medium can lack boundaries:
*Friend to me means someone that you choose to remain in contact with, not someone who can impose their friendship on you.*(Participant 1)

While the participants expressed some concerns around social technology use, they still felt it was important to adopt this technology, and they described feeling a sense of urgency around getting up to speed. They were motivated by a fear of becoming isolated in older age due to physical decline, and viewed online communication as important to their participation in the world moving forward. While they could see the potential of technology in filling this gap, the rapidly evolving nature of technology means there is a steep learning curve which could be a barrier for older people to get involved:
*But it does mean that they probably need someone from an aged care facility to come and show them how to use it, because they can’t get out to go to some kind of session. And even if they could, they’d probably hate it. Because it needs to start with how do you turn the machine on. I think there’s a lot of people that don’t realise, people out there who have never turned a computer on. It’s very frightening. They need to set up the firewalls for them with automatic updates and that kind of thing, because they don’t know about that kind of thing.*(Participant 1)

The quote emphasizes the perceived need for support in technology use. At the same time, it acknowledges a substantial set of barriers that need to be overcome, further adding to the hesitation of older community members using technologies. The participant points to the need for adequate training by people who can empathize with the low level of experience of some older adults, and who are able to engage older adults in positive ways. It is possible that, if training support is not on-site, participants might not be able to access technology due to mobility issues.

### 3.2. Motivation to Use Social Media

Participants who took part in the stage one interviews reported that their use of technology was strongly tied to the community, especially community Facebook pages, where they could campaign about local issues. One participant reported that her passion for community issues and local politics was the catalyst for her to learn how to use social media because she did not want to miss out on the conversation, while another participant noted that her online interactions on community Facebook pages often facilitated face-to-face meetings with local people. The participants reported that they would feel lonely and isolated without this technology, and recognised that for people who have difficulty leaving their homes, such as carers, social media is a powerful antidote to loneliness:
*It’s a brilliant medium for carers. My husband has Alzheimer’s and so there’s a lot of friendships that I can’t fly off to Tassie, I can’t fly off to New Zealand to see my sister, but I can keep in touch with them through social mediums and emails and things, which is very comforting, because you can feel quite isolated in that scenario … It’s huge for me. Because [my husband is] in bed by 8 o’clock at night, so we don’t go out at night much at all. Which doesn’t really worry me. But if I didn’t have that sometimes, just to send an email off, it would be very lonely.*(Participant 1)

As well as helping to combat feelings of loneliness, participants reported that technologies, such as Facebook and FaceTime, allowed them to maintain contact with family living interstate and overseas, and enhanced the quality of their social connections. In particular, one participant expressed that she has closer and more frequent contact with her granddaughter now that she has learned how to use social media:
*I’m surprised how little I miss my granddaughter, considering she’s been gone nearly a year, I suppose. Because I talk to her all the time, it’s not as though she’s gone anywhere.*(Participant 2)

This highlights the potential for social media to facilitate intergenerational communication and strengthen bonds across the life span. Furthermore, one participant reported that the mode of communication (writing or typing a message, instead of speaking) impacted the quality of communication with her son:
*Sometimes I think the communication is actually enhanced because you think about what you write a little bit more than you think about what you say. And so you can phrase things really carefully, which in my case, which is sort of like verbal diarrhoea, you know. I think in some ways it’s enhanced my relationship with my son.*(Participant 1)

### 3.3. Potential for Older People to Make Community Contributions via Social Media

As discussed previously, participants were aware of the rapidly evolving nature of social technology and felt that while the constant learning curve required commitment to ongoing learning, they had the time and interest to do this. They proposed that aged-care providers could be an excellent conduit for introducing social technology to older people, and that informal mentorship in the home or other relaxed social spaces, rather than attending specific training sessions, would increase accessibility for older people. The majority of participants were keen to learn more, but emphasised that the learning environment was important and should be informal and relaxed. One participant saw herself in the role of providing peer-to-peer technology support, emphasizing the importance of providing older adults with a low threshold to ask for help in technology use:
*We’ve put our names down for that Old Colonists Association for the retirement village. And I’ve always thought, if I get there, I could always help other people with using the technology, because I think a lot of them would like to, but they need someone they can just go to and say, what happened?*(Participant 1)

The participants commented on assumptions made about older people and social media and reported feeling undervalued in this online community space:
*I do really think that older people in the community are undervalued when it comes to things like that with social media, because I think the general opinion is, we know nothing about it, we can’t master it, we can’t navigate our way around technology. But I think that we could be a great community asset for any campaign, any local campaign like we had, because we’re home a lot.*(Participant 2)

This comment reveals that older adults are an under-tapped resource within the community due to pre-conceived notions around the types of people who are skilled at engaging with social technologies. Our participants expressed a desire to feel more valued for their skills and contributions to community life within the social media space.

### 3.4. Role of Technology in the Relationship between Place and Identity

A crucial theme to emerge from the stage two community mapping focus group was the intense relationship between place and identity, and how this was affected by a participant’s engagement with technology in their everyday lives. By drawing personal ‘maps’ of their social community located within the Heritage Village context, participants traced the contours of social relationships, meaning and connection, including their sense of space and time, over the physical landscape. This gave rise to a surprising variety of maps, including geographical, conceptual, mind maps, or lists. Some participants drew roads, roundabouts, and the houses of friends, and plotted these in relation to their own location in the village. Others drew more conceptual maps, showing social connections to people who lived both within and outside the village. In these instances, social connections were also noted with the frequency of times visited (once a week, monthly, yearly and so on), and this was accompanied by the method of contact, such as telephone, social media or physical journeys outside of the village.

The use of technology within the context of social networks seemed to expand the concept of both place and personal identity. This can be clearly seen in the map drawn by participant three (see Figure 1), who drew a large loose map of Australia on the A3 page, noting the places of most significance to him—Melbourne, Adelaide and Perth. Alongside these he drew the locations within Melbourne and the frequency of visits to each suburb, and then a reference to Fremantle and noted his yearly visits to that location. Locations and times were the major themes, frequency of visits was linked to the significance of places, or possibly, places grew in significance because of the amount of time listed there. After that was a list of reasons why each was listed—family, friends, daughter number 1, daughter number 2, etc., or activities such as shopping, and health care visits.

This participant also divided this ‘map’ into three strata’s or ‘rims’. Community rim 1 was Melbourne, Adelaide, Perth. Community rim 2 was family heritage research in Europe—Germany, England, USA—where technology became the central means by which he could search for identity and his sense of place in the world. The participant used the internet, emails, and his iPhone, dedicating about six hours per week to this pursuit.

It is interesting to note that this was cited as a second rim of community, the importance of finding heritage and meaning in locations and culture, while community rim 3 was virtual—the participant saw the boundaries defined through a digital lens. He used Facebook to connect on personal levels, find family and friends, and interest groups, and to conduct further family heritage research. This participant had a sophisticated use of social media to expand his world view and sense of place and identity, from using technology for connecting world-wide, to using apps such as Uber to visit friends and family locally.

While revealing information about social connections, the maps also provided deep insight into how people organise their social worlds. Some participants indicated a clear social hierarchy by defining their spouse and/or children as their ‘inner circle’. Sometimes, participants placed themselves at the centre of the map, at the centre of their physical and social worlds, with people and places fanning out from this geographical and psychological centre. Other participants placed themselves at the side of the map, choosing to emphasise the strong and significant social connections that dominated their perception of place and identity. However, connection to place was also linked to practical concerns, such as accessibility to health care services. One participant chose to move to the community in her retirement due to its rural feel and relative closeness and access to quality health services both within the community and in Melbourne. During the stage one community interviews, Participant 1 stated:
*I’m originally from London. When I met my husband, he was living in [a green suburb of Melbourne], but we went to live in Tasmania. And we came back eight years ago and bought a house here ready for retirement, because … we loved Tasmania, it’s a beautiful place. But the services are appalling. To the point where my husband suffers psoriasis, so he wanted to go back to a specialist he’d already seen and was told he’d have to wait a year for an appointment. So we decided to move on the urban fringe because we could get the benefits of having a rural setting, but with the services that an urban environment provides. So this was on the train line to the [anonymised hospital], has a really good doctors’ surgery, has good dentists, all that kind of thing. So it was about as far out as we could get and still get those sorts of services, and we liked it.*(Participant 1)

Another participant (Participant 4), used a Venn diagram showing his significant others (wife, daughter) overlapping with him at the centre of the page under which was written their home location (see Figure 3). From this point of reference, other geographical points fanned out into modes by which he travelled to them: car, walk, Facebook, phone, Gmail, etc. It was interesting to see that modes of transport, including walking, were considered types of technology, and that these were considered as an important means by which to reach destinations or people who were significant to him. We can see that technology is often not considered an end in itself, but rather, a means by which participants build their sense of place, identity, social connectivity and communication with the world, and that this was the primary motivation for using technology for the residents of the Heritage Village.

## 4. Discussion

With a wide body of literature already describing the benefits of social technology use for older adults (e.g., [25]), our study sought to address perceptions and attitudes amongst older adults in an urban fringe context to provide a grounding for future co-created social technology initiatives to take place in the community from an informed perspective. This research reaffirms the notion that older people perceive technology as being an important vehicle for establishing and maintaining social connections, a sense of place and identity, and for enabling contribution to issues of community importance [43]. Significantly, it also confirms previous research that has identified the supporting role technology can play in reducing social isolation in the lives of older adults [44], as well as the challenges some older adults experience when interacting via this medium. Furthermore, it highlights that the social networks of older adults are rich, complex and highly individual, revealing the need to avoid a one-size-fits all approach to technology initiatives for older adults. We argue that the disparities and complexities around technology use in this cohort call for customised, community-led social technology initiatives to be developed.

Our findings illustrate that there are differences in the way that physical and virtual spaces are mentally constructed. Technology enables people to create new spaces that are not defined geographically, but instead are defined through meaning and social connections. In particular, the concept of hierarchical circles came up through our findings. The hierarchy ensured that technology was present for some of our participants (in particular with overseas connections) in everyday life and contributed to a feeling of connection. However, participants have also built important local connections online, contributing to their feelings of connectedness to the community and place. This is different to [43] research findings, where local connectedness through technology was largely absent from everyday life. However, similar to our findings, they advocate for careful planning and consideration of the local context when aiming to improve social connections for older adults in communities [43]. While many of the participants displayed an affinity for the geographical location of Heritage Village through their drawings, and through the way they described campaigning about community issues, it was very clear that this was not a limiting social boundary. A number of the participants had emigrated to Australia from overseas and still maintain a very strong connection to their roots—this was maintained largely through use of technology to correspond with family and friends back home, to conduct research about their heritage, and even to watch the news in their first language. Furthermore, this study revealed that community participation was a very important part of the participants’ identities, and that technology was used to maintain these community links, taking them from the physical to the virtual space. We also discovered that participants had a strong drive to use technology for agency, and were using technology to contribute to the community in useful ways such as political engagement on community matters, and planning for peer-to-peer technology support in a residential setting. We were able to observe that individuals do not want to connect to technology only for their own sake, but also for the sake of meaningful contribution to a community. Technology was most valuable to our participants in maintaining connection to people, places and past histories.

While this research addresses social technology, it is worth noting that our method of engaging with participants used a low-cost approach of sticky notes, paper and pens. This process is valuable because it facilitates data collection from users that may be overlooked when high-tech approaches are used, it can be implemented in communities quickly, it uses familiar materials, and it reveals important insights into community life. Furthermore, the visualisation of qualitative and quantitative data gathered from the participant’s maps enables patterns, connections and gaps to be *seen*, which can directly inform strategic policy and service development. The value lies in making the participant’s tacit knowledge visible to a wider audience. The purpose of the map is to highlight the physical and virtual connections between people and places, ultimately revealing the extent and complexity of the social network, both within and beyond the physical boundaries of the community. Leaving the type of ‘map’ open allowed people to express themselves in an organisational way that resonated best with them at that particular point in time. The community mapping method proved to be a successful method to generate data, helping us better understand how our participants perceive place and social connections.

When considering how socialisation opportunities can be improved for older adults living in more geographically isolated locations, this leads us to think about what community technologies and infrastructure might look like. Our findings show that in order to increase social and community participation, adequate and regular technology support and agency must be given to older adults. While our study has shown that older adults have rich social lives, and that they see technology as a valuable asset in maintaining connections to people, places, and past histories, we see an opportunity to better connect technologies to actual physical space to make the physical environment and social community layer more interwoven [45]. We are sensitive in our aim to co-create an initiative that fits into an *existing* community tapestry and harnesses what is already in place. We believe that the best way forward is to co-create an initiative that connects to the community to create opportunities for social activity that are self-regulating and sustainable.

Social isolation can result from many different factors. However, living on the urban fringe or beyond means that longer geographic distances to public services and social connections can lead to isolation and decreased access to community life, negatively impacting wellbeing. There are many circumstances where isolation can suddenly be created. COVID-19 is an example of this. The social isolation that many people around the world have experienced due to measures that have been enforced to curb the spread of the virus may have some parallels with what a socially isolated person or a person with mobility issues experiences every day. While research has shown that the pandemic had a massive impact on social isolation [21], it also accelerated and increased a wider take-up of digital technologies by older adults and other vulnerable user groups (e.g., [46]). With this broad social experience of isolation, there are opportunities for designers and researchers to empathise and create more inclusive communities. With the pandemic persisting across the world, we need to consider that face-to-face interaction may continue to be limited, placing even more importance on how technology can be used to foster social connections virtually.

Based on our findings of this exploratory study, we suggest the following next steps for research. Due to the limitations of our study sample, we plan on widening our research to other natural community groups, such as the “yarn bombing” group, whose members create decorations for the village made from wool, and a book club group. This will also achieve a more representative gender balance. In order to better understand the differences of face-to-face and online social connections, we also plan to work with members of the general local Facebook group and the online group discussing planning proposals by the council for ‘Heritage Village’. Future research will also focus on a wider reach of not yet well-connected community members and their goals and needs for integration. A larger and more heterogenous participant group will provide more in-depth understanding on what constitutes social networks. Through a differentiation between face-to-face groups and online groups, we will better understand their relationship to place and community members identity towards this place. This knowledge can then be used for more effective city planning, bringing together technology-enhanced built environments that foster age-friendly and socially connected communities [47,48].

## 5. Conclusions

Positive responses to technology in our study indicate that it can be an effective support mechanism for older adult’s wellbeing, enabling them to maintain an active and meaningful role in community life. Technology was perceived as valuable because it enabled participants to maintain meaningful connections to people, places and past histories. However, we also uncovered significant concerns and fears relating to technology use. The concept of online friendship was approached with caution by some participants, and the effort required to master technology felt like a steep learning curve for some. We know that the social connections and attitudes to technology of older adults is both complex and rich. This is significant because it indicates that we need to avoid a ‘one-size-fits-all’ approach to technology initiatives for older adults, and that tailored, customised, flexible solutions need to be explored. There are opportunities for designers and researchers to work with older adults in creating more inclusive technology-connected communities. Further research is needed to explore possibilities around supporting, enhancing or replicating social connections with technology in order to make a positive difference in the lives of older adults, and to address the gap in the literature around effective technological interventions for older users in their communities.

## Figures and Tables

**Figure 1 ijerph-19-08500-f001:**
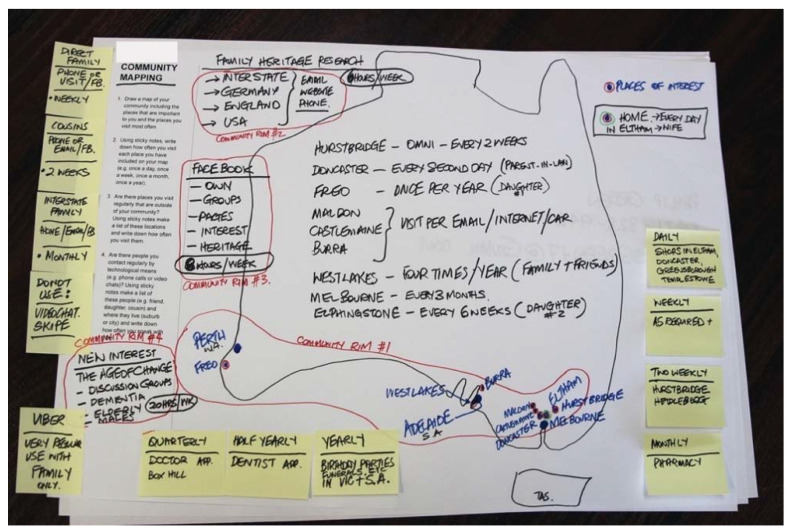
Participant map demonstrating the concept of community rims.

**Figure 2 ijerph-19-08500-f002:**
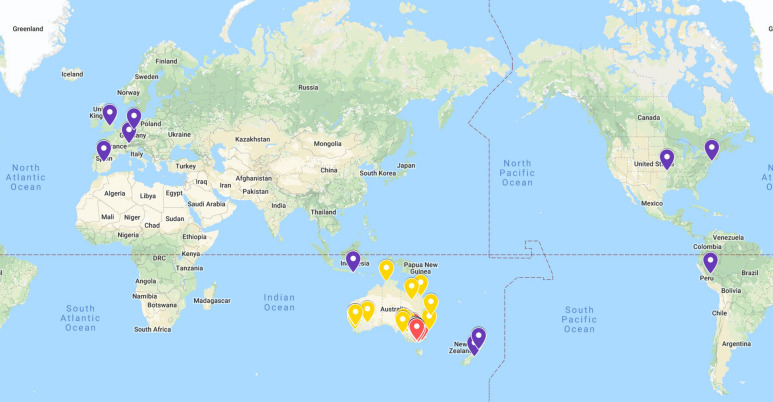
*Google My Map* collective community rim.

**Figure 3 ijerph-19-08500-f003:**
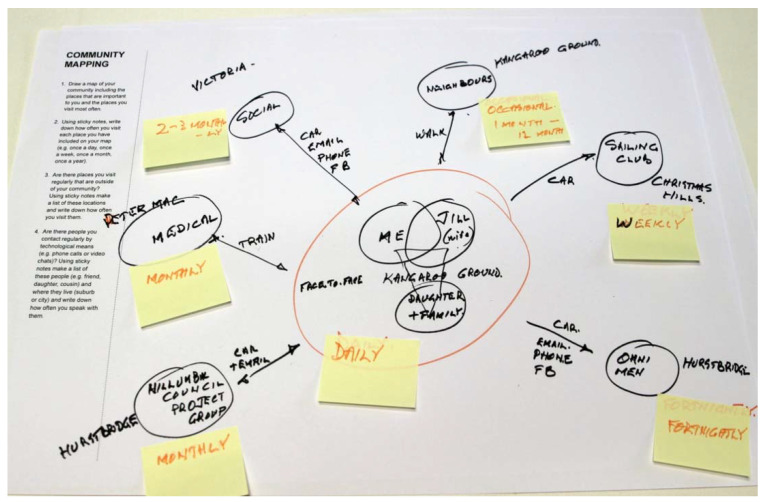
Overlapping lives in a Venn diagram (Participant 4) reveals the interwoven nature of place, identity and the use of technology as a means to communicate and build social networks.

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
