# Peer review of "Weaving Social Connectivity into the Community Fabric: Exploring Older Adult’s Relationships to Technology and Place"

_ijerph, 2022, doi:10.3390/ijerph19148500_

Round 1

Reviewer 1 Report

I’d like to thank the author(s) for this submission to IJERPH. Your research addresses an important topic with the examination of older individuals who use technology to develop and maintain social connections with others. After reading your manuscript, I offer the following suggestions:

Clearly state the purpose of the study. We as readers can infer what you wanted to do with the study, but I believe it’s important to state the actual purpose. I would also suggest including your research question(s) which helps in understanding how your research method follows from this and helps to address the purpose and question(s).

Consider including additional interviews. Your study focuses on a small population, which naturally leads to a small group from which to collect data. However, I question the use of two individuals for interviews, particularly when those individuals identify as women, while using 14 people for the focus group, and those individuals all identify as men and hail from a “men’s only social group.” Provide more rationale for these decisions.

Indicate how you reached saturation with your data collection. If you only interviewed two people, how did you determine that you reached saturation? Particularly if did not interview any men and the questions you asked the men in their focus group were not the same as the questions you asked in the interviews. The potential exists for men to have responded differently and for additional themes to emerge with their discussion of technology and social connections. 

Explain the decision to focus on this social group for the data collection process. You discuss the limited ability to generalize—which is natural for qualitative studies—but I believe you face greater limitations with this decision, unless your study purpose was centered specifically on men, which this study was not. You make the note about your desire to conduct additional research and broaden the scope and sample in future studies, but I think you would improve the current study by doing this now.

Provide additional explanation for why 14 men were included in one focus group as opposed to breaking this group into smaller groups. We typically see focus groups with no more than six people to give all participants the opportunity to speak and to ensure greater opportunities for probing, which can prove more challenging in larger sized focus groups.

You offer good findings. I would feel more comfortable with these results if I had a greater level of comfort with the data collection and sample.

Include information about the trustworthiness of the data. How did you ensure this, speaking about factors such as credibility, transferability, dependability, and confirmability. Having a statement of positionality might also be useful here.

Draw greater connections between your findings and previous research. Your literature review does not cite a specific conceptual or theoretical framework. This might be helpful when identifying theoretical implications. Without this framework, you can still compare and contrast how your findings relate to the studies you cited previously. I think this is important to show how your work advances research in this area—and provides more credibility as you move to extend the findings with your future research.

Reviewer 2 Report

The article´s topic is overall interesting. However, several things should be improved for publication.

The main issues detected are:

1. The number of people interviewed (n=2) is too low, even if only to collect life stories.

2. The fact of gender balance (interviews 2 women and large group 14 men) is an additional hindering factor to acknowledge the results.

3. Transcriptions of participants comments should be in italic or commas to be easily understandable.

4. More references within the last 3/4 years max should be included, namely to reflect the eff3ects of the pandemic. In the text throughout some sections there are several references of 2014, which is in my opinion nut sufficiently updated / current.

Reviewer 3 Report

I believe that the small sample size used in this study – even though the research is qualitative – makes accepting the article a borderline proposition. However, the topic is very relevant and the community mapping techniques innovative. I am thus inclined to recommend accepting the study with revisions. The highest priority revision is to recommend specific next steps for research given that the study is an exploratory one. The lack of next steps in this area is particularly glaring.

In the Abstract:

  • The sentence on lines 17-18 (“The primary themes that emerged from the analysis were hesitation, motivation, contribution, and place and identity.”) is too vague and should be deleted.

In the Introduction section:

  • The term “Urban fringe” should be defined when it is first introduced in line 33.
  • The statements in lines 33-34 (“Urban fringe areas in Australia are often perceived as serene, rustic places populated by people who value self-sufficiency and community spirit”), 36-38 (“we know that relationships amongst community members are complex and that a one-size-fits-all description of social values only serves to generalize or reduce what is a very rich tapestry of interwoven elements”), and 41-42 (“With increasing numbers of people choosing to live outside of urban areas as they grow older”) all need sources to back them up.
  • The sentence in lines 110-113 (“As well as technologies that address the following domains: Family, interface, privacy, phots and media, multimodal, direct communication, knowing new people, personalization and adaptation, grouping, tangible value, offline, gender, and reciprocity”) is too vague and should be deleted. In addition, it is a run-on sentence.
  • In line 121, it is unclear why the connections are necessarily more “age-friendly”. Either the term should be left out or the reason it is being used should be explained better.

In the Methods section:

  • The term “snowball sampling” in line 137 should be defined.
  • When the sample is defined in lines 137-140, one should specify how many people were natives of Australia and how many had emigrated from other countries. My reason for suggesting this addition is that the issue is brought up later in the discussion.

In the Findings section:

  • The sub-headings are very vague and should be defined better, i.e., “Hesitation about online friendships and social technology” instead of just “Hesitation”, “Motivation to use social media” instead of just “Motivation”, “Potential for older people to make community contributions via social media” instead of just “Contribution”, and “Role of technology in the relationship between place and identity” instead of just “Place and Identity”.
  • “Community rim 3” needs to be defined before the sentence in lines 313-316. Also, the sentence itself is confusing and needs to be made clearer.
  • At least two more examples of different types of social connections among the participants should be added after line 352 to give the section more depth.

In the Discussion section:

  • As already mentioned, the discussion needs to include some specific proposals for future research. This addition is critical given the study is an exploratory one.
  • The statements in lines 408-414 about how socialization opportunities can be improved for older adults living in more geographically isolated locations are too general. Recommendations for more specific actions are needed, including how to overcome the hesitations about online friendships and social technology.
  • The statement in lines 383-386 that participants were providing peer-to-peer technology support and political engagement on community matters should have been mentioned in the findings instead of first being introduced in the discussion.
  • The term “lo-fi” used in line 391 should be defined.

Reviewer 4 Report

This study makes a solid contribution. It provides some exploratory findings for an important topic. I do think there is a sampling effect that needs to be considered. Namely, those who are willing and able to participate in a focus group that was formed through recruitment of "well-connected" individuals doesn't seem like it would be generalizable to those with less social capital. It seems that those with less social capital might face greater obstacles to technology use. Likewise, the section on hesitation could be more developed. In terms of revisions, I think that the authors should give more consideration to the limitation posed by the sampling method and, if possible, develop the presentation of the hesitation section. Since this is an exploratory study, I think it would be useful for the authors to discuss what they hope to accomplish in future stages of the research. 

Reviewer 5 Report

The authors touched upon an important topic and the overall quality of the research and presentation is high. The introduction is concise and to the point, the literature review is appropriate, and the results are well-presented. The paper is close to a publishable form.

With that said, I would still like to point out some improvements for the authors to make just to substantiate the readability and contribution of this paper.

The biggest concern for me when reading this paper is the missing link between the results from the mapping exercise and the discussion. While the two interviewees could not deliver an unbiased survey, somehow the other participants in the mapping exercise should provide more information, or information to be interpreted by the author to complement the interviews. For example, the authors observed that there are hierarchical circles among the participants when they are describing their social networks. What are the implications, contributions, and inspirations for social technology during the pandemic? I did not see the authors discuss such issues, but only highlight the maintenance of their inter-continental relationships. I think the hierarchical social relationship circle has potential that has not been discussed. 

Secondly, I would like to know the authors' conception of the elderlies' social network - what does it mean to elderlies? I believe human needs elements beyond a physiologically healthy body to live and age, then what values do social networks bring to them? How is it different from the social network constructed by social technology? Since you are discussing the "place" created by social networks, you need to also discuss the cultural, social, and life history environments that together construct the place. Therefore, a conceptualization of "what is a social network" and "social network via social technology" seems necessary as a substantiating element. Perhaps this could have been discovered via the authors' survey in the earlier stage, but since the research is completed, it is still a point worth mentioning in the manuscript. Otherwise, all the discussions on "what kind" of social networks do not answer the "how" question, which inevitably renders the discussion descriptive. 

At the same time, they also sustain inter-continental social relationships via social technology. How are these related to, for example, their hierarchical circles of social networks? How does social technology perform in such a milieu? Do they only interact via social media? Or social media serves only as a platform for them to bear the hope of being able to travel and meet? After the pandemic, does this conception of social technology change?  These are also important questions for the authors to consider, which could also highlight more critical contributions to how could social technology serve the aging population. 

Round 2

Reviewer 1 Report

None

Reviewer 2 Report

well done with the alterations!

Reviewer 3 Report

In the abstract:

·       Need grammatical changes in lines 21-23: Every community is different; therefore, a tailored, customised, community-led approach to technology initiatives needs to be adopted to create more inclusive communities that are connected by technologies.

In the method:

·       Need grammatical changes in lines 152-154: Study participants (n=16) were recruited through community contacts and snowball sampling, a method in which participants recruit other participants (Goodman 2061).

·       When the sample is defined in lines 154-156, one should specify how many people were natives of Australia and how many had emigrated from other countries. My reason for suggesting this addition is that the issue is brought up later in the discussion.

Reviewer 5 Report

I think the authors have successfuly incorporated my suggestions into their new version of manuscript. 
